# Synthesis of MgO Coating Gd₂O₃ Nanopowders for Consolidating Gd₂O₃-MgO Nanocomposite with Homogenous Phase Domain Distribution and High Mid-Infrared Transparency

**Nan Wu** [1,*]**, Zhongchao Fu** [1]**, Haibo Long** [1]**, Jianming Wang** [2]**, Jun Zhang** [3]**, Zhaoxia Hou** [1]**, Xiaodong Li** [4] **and Xudong Sun** [4]

1. Liaoning Province Key Laboratory of Micro-Nano Materials Research and Development, School of Mechanical Engineering, Shenyang University, Shenyang 110044, China
2. Key Laboratory of Advanced Materials & Preparation Technology of Liaoning Province, Shenyang University, Shenyang 110044, China
3. Key Laboratory of Research and Application of Multiple Hard Films, College of Mechanical Engineering, Shenyang University, Shenyang 110044, China
4. Key Laboratory for Anisotropy and Texture of Materials (Ministry of Education), Northeastern University, Shenyang 110819, China
* Correspondence: wunan20102010@163.com; Tel.: +86-024-6226-9802

**Abstract:** Improved optical and mechanical properties are required for future infrared windows working in harsher mechanical and thermal environments than today. Ameliorating the homogeneity of the phase domain and reducing the size of the phase domain are effective approaches for enhancing the optical transmittance and mechanical hardness of a nanocomposite. In this work, we reported that the Gd₂O₃-MgO nanopowders were prepared by two different processes. The core–shell nanopowders synthesized by urea precipitation have a much lower agglomeration than the nanopowders prepared by sol–gel. Excellent transmittance (70.0%–84.1%) at 3–6 μm mid-infrared wave range and a high Vickers hardness value (10.3 ± 0.6 GPa) were maintained using the nanopowders synthesized by urea precipitation mainly due to its homogenous phase domain distribution.

**Keywords:** Gd₂O₃-MgO; core–shell; urea precipitation; nanocomposite; homogenous phase domain distribution

## 1. Introduction

Transparent polycrystalline ceramics have attracted widespread attention due to their wide range of applications such as laser hosts, infrared windows/domes, and transparent armors, instead of their single-crystal counterparts, mainly due to their processing flexibility and low cost for fabricating items with large sizes and complex shapes [1–3]. In particular, transparent polycrystalline ceramics have great potential in the application of infrared windows because of their excellent optical and mechanical properties [4–6]. With the increasingly harsh service environment of infrared windows, higher requirements are being suggested for the optical and mechanical properties of infrared windows in extreme environments [7,8]. Although traditional single-phase infrared transparent ceramics such as Y₂O₃, MgF₂, and MgAl₂O₄ have high infrared transmittance, the inevitable grain coarsening during the preparation process results in deteriorated mechanical properties, thus limiting their widespread application [5,9,10].

One way to improve the mechanical properties of the infrared transparent ceramic is to introduce a second ceramic phase to forming composites. Composites including fiber-reinforced composites [11], sandwich planet composites [12–14], composite ceramics [15], and polymers [16] are being applied in many fields. For example, in the field of mid-infrared windows, second-phase MgO is being introduced into Y₂O₃ to fabricate the

$Y_2O_3$-MgO nanocomposite with better optical and mechanical properties than any single-phase polycrystalline ceramics [17]. $Y_2O_3$ and MgO (based on the volume ratio of 50:50) were evenly mixed in order to mitigate grain growth during consolidation due to the fact that the grains of each phase pin the boundary in the other phase, thus restraining the grain boundary migration and grain coarsening. The reduced grain size will not only increase the infrared transmittance of the nanocomposite owing to the reduction in light scattering but also improve the mechanical strength according to Hall–Petch behavior [18]. Another example is via the introduction of MgO to consolidate $Gd_2O_3$–MgO nanocomposites with varied crystallographic modifications of the $Gd_2O_3$ constituent, and the nanocomposite has excellent optical and mechanical properties for mid-infrared window applications in our previous works [19,20].

The successful fabrication of nanocomposites with homogeneous phase domain distribution, fine grain sizes, and phase domain sizes is particularly challenging. Muoto et al. reported that the particle size and phase domain homogeneity of the initial nanocomposite powders directly determine the grain size and phase domain uniformity of the sintered nanocomposite ceramics, thus influencing their optical and mechanical properties [18]. Therefore, it is significant to synthesize nanocomposite powders with good dispersion and homogenous constituent phase distribution. Among the varied methods for producing nanopowders, such as spray pyrolysis [21,22], glycine–nitrate process [23,24], sol–gel combustion [24], and hydrothermal method [25], the sol–gel combustion method has always been favored, mainly due to its reduced synthesis temperature and the atomic-level mixing of starting reactants [8,17,26]. However, the properties of nanocomposite powders prepared by sol–gel combustion method such as particle size and dispersion are influenced by various factors such as starting reactants, fuel type, and equivalence ratio $\Phi$. Additionally, the sol–gel process has long powder preparation cycles which are not conductive to batch production. Urea precipitation is a powder synthesis method with a simple operation, uniform system, and controllable precipitation process [27,28]. The introduction of $Gd_2O_3$ nanoparticles as the core can induce the $Mg^{2+}$ precursor to precipitate on the surface of the nanoparticles via heterogeneous formation, thereby forming $Gd_2O_3$-MgO core–shell nanopowders. This can not only inhibit the grain overgrowth during the nanocomposite sintering process, but also solve the difficult problem of the complex and difficult-to-control precipitation variables of double cations.

In this work, to achieve the nanocomposites with homogeneous phase domain distribution, urea precipitation and sol–gel were used to synthesize $Gd_2O_3$-MgO nanopowders. We show that the core–shell nanocomposite powder with good dispersion can be obtained via the urea precipitation method, and the nanocomposites with homogenous distribution of constituent phases, higher optical transmittance, and a hardness value can be obtained after hot-press sintering. The underlying influence mechanism of the nanopowders with different morphologies on the microstructure of the nanocomposites was discussed.

## 2. Material and Methods

### 2.1. Preparation of the $Gd_2O_3$-MgO Core–Shell Nanopowders by Urea Precipitation

MgO coating $Gd_2O_3$ core–shell nanopowders were synthesized by urea precipitation. The raw materials were gadolinium oxide nanopowders (5N), urea, and magnesium nitrate hexahydrate ($Mg(NO_3)_2 \cdot 6H_2O$). All the raw materials were of analytical grade (Sinopharm Chemical Reagent, Shanghai, China). First, the 74.78 mmol $Mg(NO_3)_2$ and stoichiometric urea were dissolved in deionized water into a three-necked flask to form a mixed solution. Then, 6.99 g $Gd_2O_3$ was weighted with a volume ratio of 50:50 to MgO (monoclinic-$Gd_2O_3$:cubic-MgO) into the container via sufficient stirring and dispersing. After that, the dispersed $Gd_2O_3$ was put into the above solution in the three-necked flask. Additionally, then, the solution in the three-necked flask was heated and stirred at 90 °C for 3 h to obtain the turbid solution. After that, the resulting suspension was obtained by filtration and then the suspension was put into a 90 °C oven for drying, thus obtaining the dried precursors. Then, the dried precursors were calcined at 850 °C to prepare the core–shell nanopowders.

The nanopowders were ball-milled with $Al_2O_3$ balls in a roller ball mill (100 r/min). The slurry was placed into an 80 °C oven for drying. Finally, the nanocomposite powders were sieved by a 200-mesh screen.

### 2.2. Synthesis of the Gd₂O₃-MgO Nanopowders by Sol–Gel Combustion

Nanopowders of Gadolinium oxide–magnesium oxide were synthesized by sol–gel combustion technique. Commercial $Gd(NO_3)_3 \cdot 6H_2O$, $Mg(NO_3)_2 \cdot 6H_2O$, $C_6H_{10}O_8$, and ethylene glycol were used as raw reactants (Sinopharm, Shanghai, China). In a typical synthesis procedure, firstly, distilled deionized water, citric acid, and glycol were mixed to form an aqueous solution. Secondly, the 0.03854 mol $Gd(NO_3)_3$ and 0.07478 mol $Mg(NO_3)_2$ solutions were added into the above aqueous solution to prepare a clear sol. The sol was placed into an oven at 90 °C to obtain the precursor. Then, the precursors were calcined at 600 °C to obtain the nanopowders. The nanopowders were also obtained via a series of ball milling, drying, and screening processes consistent with Section 2.1.

### 2.3. Sintering of the Gd₂O₃-MgO Nanocomposite

The treated (calcined, ball-milled, and screened) $Gd_2O_3$-MgO nanopowders synthesized by two methods were dry-pressed at 100 MPa in a steel-mold ($\Phi_{diam}$ = 25 mm) to obtain the green bodies, and then the green bodies were sintered via hot pressing at 1350 °C for 0.5 h. The load was applied at the temperature of 600 °C, and gradually increased to 50 MPa at 1000 °C. Post-sinter annealing was carried out at 1000 °C for 20 h in air. Both surfaces of the samples were polished, and then thermally etched for the property characterization.

### 2.4. Investigation for the Nanocomposite Powders and Sintered Ceramics

Simultaneous thermogravimetric and differential scanning calorimetry analyses of the precursor synthesized by two different processes were carried out on a TGA-DSC apparatus (STA449F3, Netzsch, Selb, Germany). The precursor to be analyzed was heated at 10 °C/min in flowing air. The XRD analysis was conducted to identify the structures of the nanopowders and nanocomposites by X-ray diffraction (X'pert, PANalytical, Almelo, The Netherlands) using Cukα radiation. The crystal size of the nanocomposite powders can be calculated by Scherrer's formula:

$$D_{XRD} = (K\lambda)/(\beta cos\theta) \tag{1}$$

where $K$ belongs to a constant, taking 0.89; $\lambda$ represents the wavelength of Cukα wavelength; the width at half height for diffraction peak of the measured sample is represented by $\beta$; $\theta$ is the Bragg diffraction angle; and $D_{hkl}$ means the crystal size of the nanopowders. The specific surface area of the nanocomposite powders was determined using a gas sorption analyzer by the Brunauer–Emmett–Teller (Tri-Star II 3020, Norcross, GA, USA) method. The mean particle size of the nanopowders was calculated via Formula (2):

$$D_{BET} = 6000/(\rho S_{BET}) \tag{2}$$

where $S_{BET}$ denotes the specific surface area of the nanopowders; $\rho$ stands for the theoretical density of the nanopowders; and $D_{BET}$ represents the mean particle size. The ratio of $D_{BET}/D_{XRD}$ was adopted to evaluate the agglomeration factor of the nanocomposite powders synthesized by a different method. Transmission electron microscopy (JEM-2100F, JEOL, Tokyo, Japan) determined the morphologies of the nanopowders prepared by different techniques. The structures and morphologies of the sintered nanocomposite ceramics were evaluated via scanning electron microscopy (JSM-7001F, JEOL, Tokyo, Japan). The statistics of the mean grain size for nanocomposite ceramics were obtained by measuring at least 200 grains on the BSE images using the line-intercept method. The infrared in-line transmittance of the nanocomposite was measured using a Fourier transform infrared spectrometer (Nicolet iS5, Thermo Scientific, Waltham, MA, USA). The test for the Vickers

hardness was performed via a tester (401 MVD, Wolpert, Norwood, MO, USA), the load was 500 g, the dwell time was 10 s, and the average of 10 measurements was taken as the final hardness value of the specimen [29].

## 3. Results and Discussion

### 3.1. The Thermal Behaviors of the Precursors Synthesized by Two Processes

Figure 1 shows the simultaneous thermal behaviors of the precursors synthesized by the two methods. For the precursor synthesized by urea precipitation (shown in Figure 1a), based on previous research results [30], in the process from room temperature to 1000 °C, the total mass loss is 41.7%, which is divided into four steps. The first stage is from room temperature to 196 °C, with a mass loss of 9.6%, and an endothermic peak at 170 °C is ascribed to the evaporation of absorbed water and the release of bound water. The mass loss in the second stage is 2.2%, which occurs in the temperature range between 196 and 281 °C. The exothermic peak at 271 °C on the DSC curve is mainly due to the decomposition and oxidation of nitrate. The third period is from 281 to 624 °C, and the mass loss is 18.3%; the endothermic peaks are located at 437, 466, and 614 °C, and are related to the decomposition of hydroxides and carbonates. The 11.6% mass loss occurs in the fourth step, and there is an exothermic peak at 629 °C which is attributed to the crystallization of oxides. No obvious weight loss was observed after 850 °C, indicating that MgO-coated $Gd_2O_3$ nanopowders can be obtained at 850 °C. Figure 2a shows that the precursor calcined at 850 °C is composed of $Gd_2O_3$ and MgO phases.

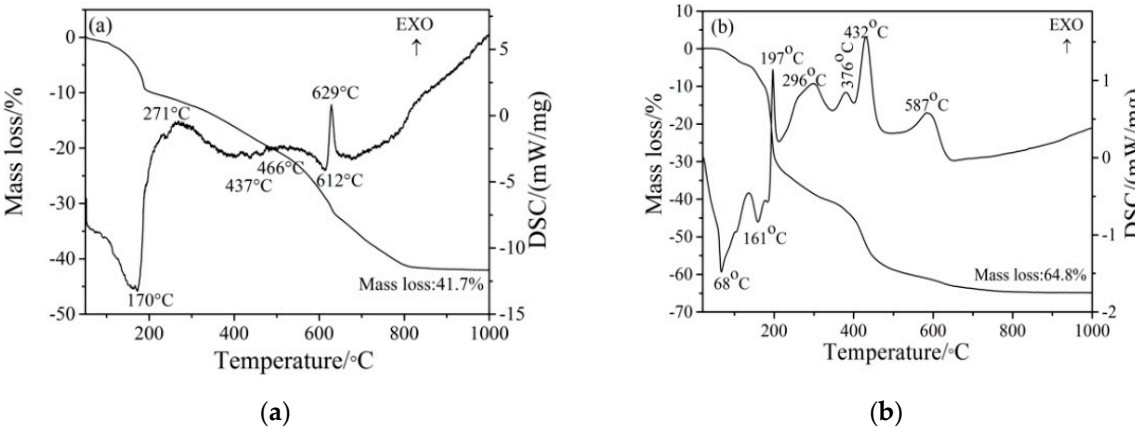

(**a**)                                                    (**b**)

**Figure 1.** Plot of the differential thermal analysis and thermogravimetry data obtained at a heating rate of 10 °C/min from the precursors synthesized by different methods: (**a**) urea precipitation; and (**b**) sol–gel.

For the precursor synthesized by sol–gel (shown in Figure 1b), it can be clearly seen that four-period weight loss occurred. The first weight loss period over the temperature ranging from room temperature to 178 °C is assigned to residual water volatilization. The DSC curve has two remarkable endothermic peaks at 68 and 161 °C. A sudden drop was observed in the second period weight loss, which was from 178 to 222 °C. In this period, there was a sharp exothermic peak at 197 °C, since the precursor was ignited. This is because there is a redox reaction between nitrate and citrate, thus generating intermediate products and releasing carbon oxide and nitrogen oxide gases [31,32]. The third period occurred between 222 and 510 °C and there are three exothermic peaks (296, 376, and 432 °C) being caused by the decomposition of the intermediate products. An exothermic peak at 587 °C emerged in the fourth period, which is ascribed to the decomposition of the remaining organics and the oxide crystallization. The XRD result in Figure 2a proves that the precursor heated at 600 °C consists of $Gd_2O_3$ and MgO. The thermal decomposition process of the precursor is consistent with our previous results [20].

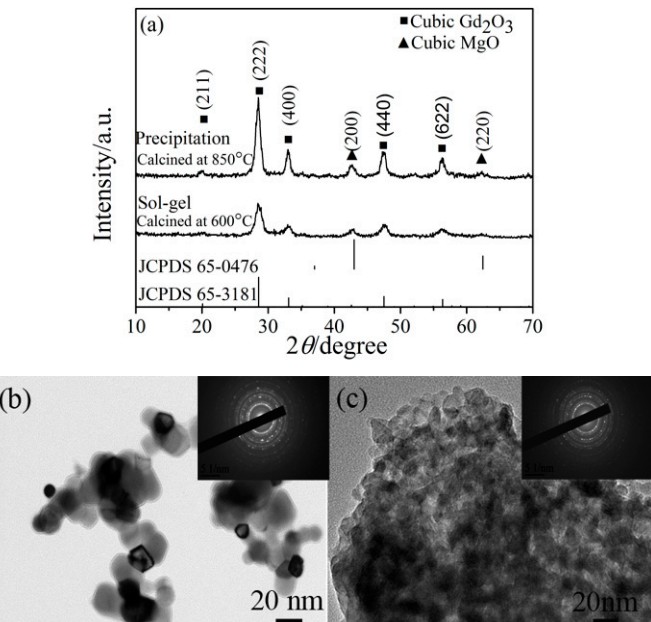

**Figure 2.** (**a**) XRD patterns of the nanocomposite powders preparation via different methods. TEM images of the calcined nanopowders synthesized by different methods; (**b**) urea precipitation; and (**c**) sol–gel. The insets are the SADPs of the calcined nanopowders synthesized by different methods.

### 3.2. Effects of Preparation Process on the Performances of Nanocomposite Powders

Figure 2a shows the XRD patterns of the nanopowders synthesized from a different method. Obviously, the composite nanopowders were cubic $Gd_2O_3$ and MgO phases, regardless of the synthesis methods. The diffraction peaks of the sample synthesized by urea precipitation are much more intense and sharper than that of the sample synthesized by sol–gel. Table 1 shows the particle size ($D_{BET}$), crystal size ($D_{XRD}$), and agglomeration factor calculated. Although the crystal sizes of the nanocomposite powders synthesized by sol–gel are finer than that of the nanopowders synthesized by urea precipitation, the agglomeration factor has an obvious increase. TEM results further verified the conjecture for the morphologies for the two nanocomposite powders synthesized by different methods. The sample synthesized by urea precipitation has a larger particle size and better dispersion than those synthesized by sol–gel. The diffuse amorphous rings in the insets selected area diffraction patterns (SADPs) are consistent with the XRD data (shown in Figure 2a) which indicate that the nanopowders synthesized by urea precipitation have better crystallinity. In addition, the sample shown in Figure 2b exhibits a clear interface between core and shell, indicating that the $Gd_2O_3$, as a core, is successfully cladded with MgO as a shell.

**Table 1.** Characterization of the $Gd_2O_3$-MgO nanocomposite powders synthesized from different methods.

| Experiment Method | $D_{BET}$ (nm) | $D_{XRD}$ (nm) | Agglomeration Factor |
|---|---|---|---|
| Precipitation | 59.8 | 32.3 | 1.9 |
| Sol–gel | 29.7 | 9.1 | 3.3 |

Clearly, the above results demonstrate that the properties of the composite nanopowders are greatly affected by the synthesis method. With the variation of the synthesis method, the crystal size and the agglomeration state for nanocomposite powders varied remarkably. The discrepancies can be related to the thermal decomposition processes of the precursors exhibited in the TG-DSC curves. For sol–gel process, the sharp exothermic peak of 197 °C in the DSC data and the sudden weight loss on the TG indicate that the decomposition process is instantaneous. The rapid reaction rate can restrict particle growth and

eventually form extremely fine crystallite sizes [18,24,33]. In the sol–gel reaction system, excessive fuel will generate a lot of heat in the later decomposition process, leading to the agglomeration of nanocomposite powder. Therefore, although the crystallite size of nano powders synthesized by sol–gel method is much finer, the agglomeration of nanopowders is more serious. In addition, Figure 1b shows that the peaks at 296, 376, and 432 °C in the DSC data also verified that the reaction at the later stage was very intense. Such a result is similar to our previous study [20]. For urea precipitation, the entire reaction process is gentle without violent decomposition and abrupt weight loss, since no extra heat is required for nanopowder agglomeration. Therefore, the nanopowder synthesized by urea precipitation has a lower agglomeration state.

### 3.3. Effects of Synthesis Process on the Phase, Microstructure, Optical, and Mechanical Properties of Nanocomposites

Figure 3 shows the structures of the $Gd_2O_3$-MgO nanocomposite ceramics. The characteristic monoclinic $Gd_2O_3$ and cubic MgO peaks emerged without any detectable impurity phase, irrespective of the two synthesis methods. It is worth noting that the $Gd_2O_3$ powders exhibited a cubic phase (shown in Figure 2a), but it turned into a monoclinic phase after sintering at 1350 °C. It is reported that $Gd_2O_3$ undergoes a cubic to monoclinic transformation above 1250 °C [34]. It is worth noting that the high-temperature monoclinic $Gd_2O_3$ is unstable in thermodynamics. However, the two samples are still retained in a monoclinic $Gd_2O_3$ phase at room temperature. The reason for the absence of the monoclinic-cubic reverse transition can be attributed to the slow atom spread dynamics and rapid cooling after hot pressing [35,36].

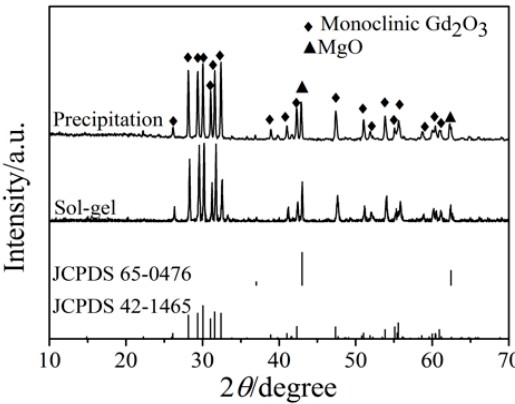

**Figure 3.** XRD patterns of the sintered nanocomposite ceramics with nanocomposite powders from different methods.

The effect of nanocomposite powders synthesized by different methods on the microstructure of the hot-pressed nanocomposites was characterized via BSE images. Figure 4 shows the representative BSE images. There is no significant difference in the grain size of the two samples; however, the structure and distribution of the phase domain are quite different. For the nanocomposite sintered using the urea precipitation nanopowders (shown in Figure 4a), the grain sizes of $Gd_2O_3$ (bright phase) and MgO (dark phase) are 160 and 120 nm, respectively, and the sample has a very even microstructure and a homogenous two-phase distribution. For the nanocomposite sintered using the sol–gel nanopowders, the grain sizes of $Gd_2O_3$ and MgO become slightly finer—140 and 130 nm. However, the clusters in each phase are large, which leads to a large-sized phase domain and inhomogeneous phase domain distribution.

One of the major goals of improving the phase domain homogeneity in synthesized nanopowder is to suppress the grain coarsening during the subsequent sintering processing of the nanopowders. It can be seen from Figures 2 and 4 and Table 1 that the agglomeration state of the initial nanopowders has a significant effect on the microstructure of the sintered

nanocomposite. The speculative schematic diagrams of the microstructure evolution process of the nanopowders prepared by two different processes during the sintering process are shown in Figure 5. For the nanopowders synthesized by urea precipitation, $Gd_2O_3$ as a core is effectively coated by MgO as shell, and the $Gd_2O_3$ grains are pinned by the MgO grains; thus, due to the core–shell structure constrains and the effective pinning effect, the microstructure will be stable until the coarsening of the one-phase domain can occur. However, the severe agglomeration of nanopowders prepared by sol–gel may cause each phase domain in the powder to contain multiple homophase particles. During the sintering process, the homophase particles in each phase domain will rapidly combine and grow. The reason for this phenomenon is that the agglomerated homophase particles only need to cross the homophase grain boundary, so the grains only undergo a short-distance rearrangement of atomic positions; eventually, resulting in large grain and phase domain sizes and inhomogeneous phase domain distribution.

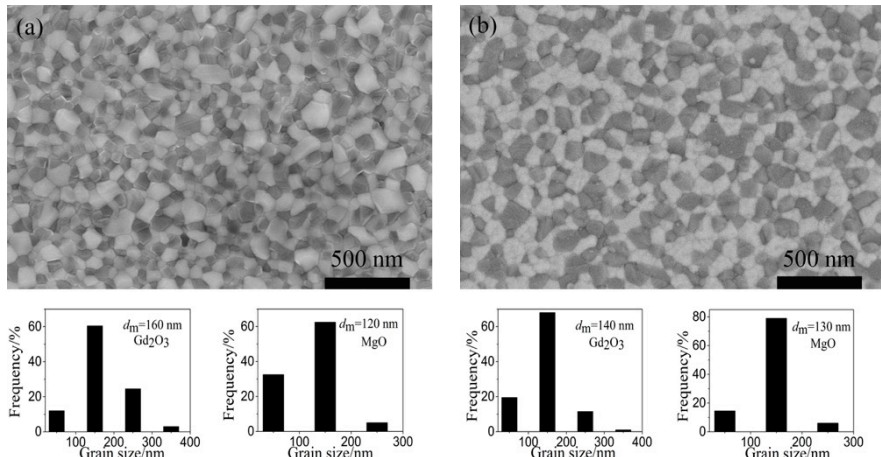

**Figure 4.** BSE images and grain size distributions of the sintered nanocomposite ceramics with nanocomposite powders with two methods: (**a**) urea precipitation; and (**b**) sol–gel.

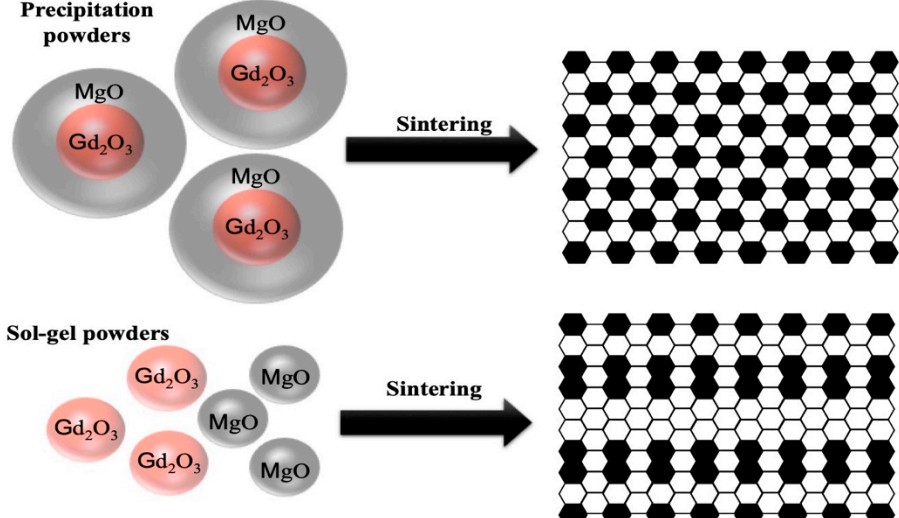

**Figure 5.** Diagrammatic sketch of the phase domain and grain size evolution during the sintering of composite materials using different nanopowders.

As can be seen from the above analysis, initial nanocomposite powders with slight agglomeration, fine phase domain size, and homogenous phase domain distribution are conductive to exert pinning effect and restrain grain growth in the subsequent sintering process. This hypothetical microstructure evolution was well verified in this work. The

agglomeration state of the core–shell nanopowders synthesized by urea precipitation is slighter than that of the nanopowders synthesized by sol–gel (as shown in Figure 2 and Table 1). After hot pressing, the homogeneity of the phase domain of the former is obviously superior to that of the latter. This phenomenon of the agglomeration state of nanocomposite powder affecting the phase domain uniformity of the nanocomposite after sintering was observed in the previous fabrication of $Y_2O_3$-MgO and MgO-$ZrO_2$ nanocomposites [18,33,37].

Figure 6 shows the optical transmittance in the infrared band of the $Gd_2O_3$-MgO nanocomposites sintered using the nanopowders synthesized by two methods. The microstructure, such as the porosity, grain size, size, and homogeneity of the phase domain greatly affects the infrared transmittance of the sample. On the one hand, the porosity is the main factor affecting the transmittance when the ceramic bulk is at a low relative density, because light scattering from a large number of pores will seriously deteriorate the light transmission because of the different refractive index of MgO, $Gd_2O_3$, and air. Additionally, the infrared transmittance also degrades when the relative density of the sample is high due to grain overgrowth and uneven phase domain distribution. As shown in Figure 4, there is no significant difference in the grain size between the two samples, and there are no obvious pores. Therefore, the distribution and uniformity of the phase domain play a dominant role in affecting the infrared transmittance in this work. As expected, the sample sintered using the core–shell nanopowders showed excellent transmittance (70.0%–84.1%) at 3–6 μm mid-infrared thanks to the more homogeneous phase domain distribution. Moreover, the two specimens have several absorption peaks at approximately 7 μm due to the asymmetrical and symmetrical stretching vibrations of the carboxylate groups, forming in the starting powders or subsequent sintering process because of the remaining carbon-containing groups. This is detrimental to the optical performance of a nanocomposite in the infrared wavelength range [38]. Therefore, the production of carbon-free nanocomposite powders with good dispersion and a uniform phase domain is key to further improving the infrared transmittance performance of nanocomposites. The detailed results will be described in the subsequent paper.

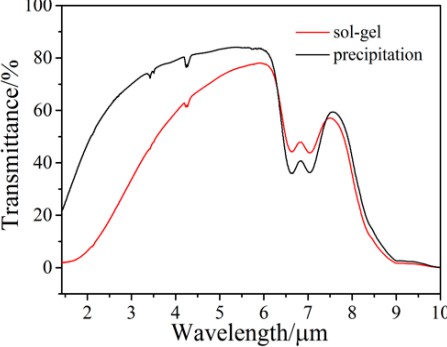

**Figure 6.** IR transmission spectra of the sintered nanocomposite ceramics with nanocomposite powders with two methods (the thickness of the sample is 2.0 mm).

The mechanical hardness of the sintered $Gd_2O_3$-MgO nanocomposites using nanopowders with different methods were measured. As Figure 7 shows, the Vickers hardness of the sample sintered using the core–shell nanopowders synthesized by urea precipitation is higher than that of the sample sintered using the nanopowders prepared by sol–gel due to a more homogeneous phase domain distribution. The hardness value of $10.3 \pm 0.6$ GPa is significantly higher than that of pure dense MgO and $Y_2O_3$ ceramics (5–7 GPa) [39,40], and the hardness value is similar to that of $Y_2O_3$-MgO reported by Xu et al. ($10.0 \pm 0.1$ GPa) [7] or Ma et al. ($10.6 \pm 0.2$ GPa) [24]. In addition, the optical and mechanical properties will be further improved for adapting to a much harsher environment in the future when the microstructure is further optimized. There is still room to homogenize the phase domain

distribution and reduce the grain size via the optimization of the nanopowders' preparation and sintering process.

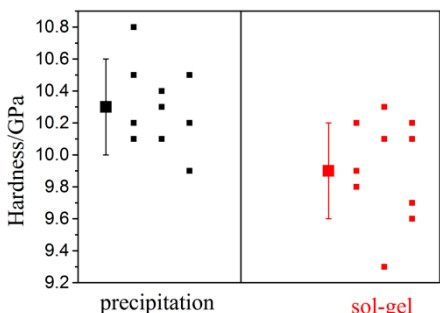

**Figure 7.** Vickers hardness of the sintered $Gd_2O_3$-MgO nanocomposites using nanopowders with different methods.

## 4. Conclusions

The $Gd_2O_3$-MgO nanopowders with different morphologies were synthesized by urea precipitation and citric sol–gel, respectively. The thermal behaviors of the precursors prepared by two processing methods influenced the agglomeration state of the nanopowders. The $Gd_2O_3$-MgO nanopowders with a core–shell structure have lower agglomeration, and are make it easy to obtain nanocomposites with a homogeneous phase domain distribution after hot-press sintering. For the $Gd_2O_3$-MgO nanocomposite sintered at 1350 °C and 50 MPa using the core–shell nanopowders, the average sizes of $Gd_2O_3$ and MgO are 160 and 120 nm, respectively. The nanocomposite with even two-phase distribution exhibits an outstanding transmission (70.0%–84.1%) in the mid-infrared range and a high hardness value (10.3 ± 0.6 GPa). The results indicate that core–shell nanopowder is conductive to restraining the growth of the phase domain size and the formation of an inhomogeneous phase domain for nanocomposites.

**Author Contributions:** Writing—original draft preparation, data curation, formal analysis, and investigation, N.W.; Data Curation, H.L. Writing—review and editing, X.L. Supervision, Z.F., J.W., J.Z., Z.H. and X.S. All authors have read and agreed to the published version of the manuscript.

**Funding:** This research was funded by postdoctoral research start-up funding (No. 12205020052022010216).

**Institutional Review Board Statement:** Not applicable.

**Informed Consent Statement:** Not applicable.

**Data Availability Statement:** Data sharing is not applicable to this article.

**Conflicts of Interest:** The authors declare that they have no known competing financial interest or personal relationships that could have appeared to influence the work reported in this paper.

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
