# Peer review of "Synthesis of MgO Coating Gd2O3 Nanopowders for Consolidating Gd2O3-MgO Nanocomposite with Homogenous Phase Domain Distribution and High Mid-Infrared Transparency"

_coatings, doi:10.3390/coatings12101435_

Round 1
Reviewer 1 Report
Comments, suggestion and questions in the attach document.

Author Response
Dear Editor and Reviewer:
Thank you very much for your letter and for the reviewer’s comments concerning our manuscript entitled “Synthesis of MgO coating Gd2O3nanopowders for consolidating Gd2O3-MgO nanocomposite with homogenous phase domain distribution and high mid-infrared transparency” (ID: Coatings-1885649). These comments are all valuable and very helpful for revising and improving our paper. We have studied the comments carefully and have made corrections which we hope meet with approval for the publication of our paper. Revised portion are marked in red in the paper. The main corrections in the paper and the responds to the reviewer’s comments are as following:
Comment 1: Description, state of the art, and in general the introduction content is clear. The reading of the section is pleasant and simple; however, there is no approach to real sectors. Results are not entirely clear because there are no strategies to compare with those currently used by the industry. What is the real application of this kind of coatings in the industry?
Reply: Thank you so much for your kind comments.
In previous works, most of the nanocomposite powders were synthesized by sol-gel method, but this process is facile to cause powder agglomeration and result in the rapid aggregation and growth of homophase particles in each phase domain, thus deteriorating the microstructure and properties of nanocomposite ceramics. In this work, we mainly provide a new synthesis method of nanocomposite powders for the fabrication of nanocomposite ceramics. The synthesized nanocomposite powder with core-shell structure can exert the pinning effect more effectively during sintering, thus improving the microstructure and properties of nanocomposite ceramics.
Comment 2: Some of the discussions about material properties are based on nanopowders morphology and size; however, no reasonable information about that is presented in the introduction. I consider that it is necessary to introduce to readers the influence of those characteristics on the abovementioned properties. Please include this information.
Reply: Thank you so much for your kind comments.
We have added relevant information in the paper.
Comment 3:This section is very clear, the methodologies and equipment are well described. However, in sections 2.1 and 2.2 appears that nanopowders were ball milled but no conditions are shown. This information is mandatory to understand how the nanopowders were obtained. I don’t know if researchers use a conventional planetary mill, a ball mill, or a high-energy ball mill. Please give more details about the ball milling process.
Reply: Thank you so much for your kind comments.
We have added relevant ball milling information in the paper.
Comment 4: In lines 151 to 162 there are a series of peaks associated with different reactions that occur at different temperatures for each method of nanopowders preparation. Despite the knowledge and expertise of the researchers no evidence of that reaction and products of that reaction are depicted. There are no thermodynamical considerations for that reactions. how does the author probe those reactions? what are the most probable reactions in each system: sol-gel and urea precipitation?
I consider it very useful to carry out an analysis of Gibbs free energy as a function of temperature in the systems studied, that is sol-gel and urea precipitation. Moreover, according to XRD and TEM results, the nanopowders tend to be amorphous and as we all know, the unstable amorphous structure features higher activity than the stable crystalline structure: for this reason is fundamental to understand the reaction presented in the systems.
According to Fig 2(a), it is possible to affirm that nanopowders are amorphous? SAED is an exceptional technique to define if your materials are amorphous or not, more when the authors present TEM images. Obtaining SAED patterns should not be a challenge if the sample nanopowders exist. Please include these results and discussion.
Reply: Thank you so much for your kind comments.
The literatures related to the thermal decomposition behavior of precursors synthesized by urea precipitation method and sol-gel method has been added to this paper. In the future, we will use Gibbs free energy to systematically analyze the reaction process according to what you said. Additionally, we are very sorry that the description of Figure 2 caption caused your misunderstanding. The two TEM images in Fig. 2 are the morphologies of the calcined nanopowders synthesized by two methods. We have corrected the Figure captions.As for the detailed behavior of each step of thermal decomposition of the precursor, we will adopt the opinions of the reviewer and discuss it in detail in the subsequent papers.
Comment 5: In lines 208-212 do you have evidence of that hypothesis? Are there investigations with similar results?
Reply: Thank you so much for your kind comments.
The effect of fuel consumption on powder agglomeration in sol-gel reaction has been systematically discussed in our previous paper. We have added this literature in this paper.
Comment 6: In lines 241 to 244 how the grain size was calculated? What technique or methodology do you use?
Reply: Thank you so much for your kind comments.
More than 200 grains were measured on the BSE image by the line-intercept method to obtain the average grain size of the nanocomposite.
Comment 7: In Fig 7. High dispersion of data is shown. what could be the source of the high dispersion of the data? The procedure of how the nano-hardness was done is not depicted and therefore could be the reason for that dispersion. Moreover, 10 measurements do not guarantee the validity of the tests. For this reason, is important to take a look at the 10 load-displacement nanoindentation curves to
provide a mechanical behavior of a material's response to contact deformation. Please provide these curves (not for the paper but for analysis aims) and analyze the results according to the Oliver- Phar method to validate the behavior and forms of loading and unloading curves. With this information, it is possible to discuss the results or repeat the tests selecting the best curves.
Reply: Thank you so much for your kind comments.
We express our great regret and regret to you. Due to the COVID-19 epidemic in Shenyang, the use of colleges or testing institutions is restricted, so we can't repeat the experiment. When the epidemic situation improves, we will adopt your opinions to conduct detailed analysis and supplement on mechanical properties.Once again, I would like to express our apologies and your help for our paper.

Reviewer 2 Report
I see this paper with the title of "Synthesis of MgO coating Gd2O3 nanopowders for consolidating Gd2O3-MgO nanocomposite with homogenous phase domain distribution and high mid-infrared transparency". This paper has a novel topic but I have the below concerns about this paper before final decision :
1-The first concern is about the interaction force between matrix and nanoparticle which is not applied in this paper.
2-The introduction section can be improved:
Waves in Random and Complex Media, (2022) 1-25, https://doi.org/10.1080/17455030.2022.2030499; Polymer Composites 43 (2022), 282-298; Mechanical Systems and Signal Processing 178 (2022) 109269; Thin-Walled Structures 178 (2022) 109495; Journal of Materials Research and Technology (2022), https://doi.org/10.1016/j.jmrt.2022.06.008.
3-Abstract gives information on the main feature of the performed study, but a couple of sentences about background of the study must be added. However, a concise abstract is needed.
4-Authors must clarify necessity of the performed research. Aims and objectives of the study, and also differences with the previous review papers must be clearly mentioned.
5-The conclusion must be more than just a summary of the manuscript. List of references must be updated based on the proposed papers. Please provide all changes by red color in the revised version.
Author Response
Dear Editor and Reviewer:
Thank you very much for your letter and for the reviewer’s comments concerning our manuscript entitled “Synthesis of MgO coating Gd2O3nanopowders for consolidating Gd2O3-MgO nanocomposite with homogenous phase domain distribution and high mid-infrared transparency” (ID: Coatings-1885649). These comments are all valuable and very helpful for revising and improving our paper. We have studied the comments carefully and have made corrections which we hope meet with approval for the publication of our paper. Revised portion are marked in red in the paper. The main corrections in the paper and the responds to the reviewer’s comments are as following:
Comment 1: The first concern is about the interaction force between matrix and nanoparticle which is not applied in this paper.
Reply: Thank you so much for your kind comments.
I’m sorry, we don’t quite understand what you mean. Do you mean the formation process of the Gd2O3-MgO core-shell nanopowders? The mechanism of the self-assembled core-shell precursors can be divided into two stages. Firstly, as urea hydrolysis began, the precipitant groups of OH-, CO32-, and HCO3-were formed and modified the surface of the nanoparticle. The Mg-compound was precoated on the surface of Gd2O3nanoparticles, driven by electrostatic attraction. Secondly, the assembly of the tiny Mg-compound particles is formed in the burst nucleation process onto Gd2O3nanoparticle surface, mainly driven by its high surface energy. Meanwhile, the size of the powder is bound by the core-shell structure, so that the grains do not grow excessively during the sintering process.
Comment 2: The introduction section can be improved
Reply: Thank you so much for your kind comments.
The introduction section has been improved according to the literatures you provided.
Comment 3: Abstract gives information on the main feature of the performed study, but a couple of sentences about background of the study must be added. However, a concise abstract is needed.
Reply: Thank you so much for your kind comments.
We have further revised the abstract.
Comment 4:Authors must clarify necessity of the performed research. Aims and objectives of the study, and also differences with the previous review papers must be clearly mentioned.
Reply: Thank you so much for your kind comments.
Our main goal is to synthesize core-shell structure nanocomposite powder for suppressing the aggregation of homophase particles during sintering, thus improving the microstructure and properties of the nanocomposites. Predecessors mainly used sol-gel combustion method to prepare nanocomposite powder, but its disadvantage is that the powder agglomeration is serious, particularly, homophase particle aggregation. To low the agglomeration of powder, it is necessary to adjust various process parameters, such as Φ Value, pH, etc. In this paper, we use a simple precipitation method to synthesize core-shell structure nanocomposite powders, which can effectively improve the agglomeration of homophase particles, thereby optimizing the microstructure of nanocomposite ceramics and improving its performance.
Comment 5: List of references must be updated based on the proposed papers. Please provide all changes by red color in the revised version.
Reply: Thank you so much for your kind comments.
We have updated the references according to your requirements, and the modified parts have been marked in red color.

Reviewer 3 Report
In this paper, the authors reported the Gd2O3-MgO Nano powders preparation with different morphologies using urea precipitation and sol-gel process. Briefly explain how the particle's behavior and phase domain change by the hot pressing process at high temperature at 1350OC.
After addressing the following, the paper's subject is suitable for publication in Coatings-MDPI.
The paper is well written, but in my opinion, the authors should add and need to discuss some more characterization results. In this way, the paper will be more interesting for the readers.
· In the experimental section, the synthesis of Nanopowders section 2.1 & 2.2, why the Nano powders calcined at 850 oC for urea precipitation method, at 600 oC for Solgel combustion method. Briefly explain the reason for maintaining two different calcination temperatures for two different methods.
· In section 2.4, give the standard of measurements for the characterization of Nanopowders using Vickers hardness and other analytical instruments. Also, introduce the references for the standards in the manuscript.
· What is the particle size obtained in both synthesis methods before ball milling? Add the XRD and surface morphology results of freshly prepared nanoparticles using two methods after calcination.
· Explain the effect of the ball milling process on the particle's grain refinement using SEM images
Author Response
Dear Editor and Reviewers:
Thank you very much for your letter and for the reviewer’s comments concerning our manuscript entitled “Synthesis of MgO coating Gd2O3nanopowders for consolidating Gd2O3-MgO nanocomposite with homogenous phase domain distribution and high mid-infrared transparency” (ID: Coatings-1885649). These comments are all valuable and very helpful for revising and improving our paper. We have studied the comments carefully and have made corrections which we hope meet with approval for the publication of our paper. Revised portion are marked in red in the paper. The main corrections in the paper and the responds to the reviewer’s comments are as following:
Comment 1: In this paper, the authors reported the Gd2O3-MgO Nano powders preparation with different morphologies using urea precipitation and sol-gel process. Briefly explain how the particle's behavior and phase domain change by the hot-pressing process at high temperature at 1350 oC.
Reply: Thank you so much for your kind comments.
For the nanopowders synthesized by the sol-gel method, the agglomeration of the nanopowders causes multiple homophase grains in each phase domain to grow rapidly, because the homophase agglomerated grains only need to cross the homophase grain boundaries, so the grains will only undergo a short-distance rearrangement of atomic positions, thus forming a new and larger grain, resulting in a large grain size and a large phase domain size of the sintered sample. However, for the nanopowders synthesized by the urea precipitation, since Gd2O3is coated by MgO, a good pinning effect is formed. The Gd2O3grains are pinned by the MgO phase and vice versa and thus, the microstructure will be stable until coarsening of the MgO phase domains can occur.
Comment 2:In the experimental section, the synthesis of Nanopowders section 2.1 & 2.2, why the Nano powders calcined at 850 oC for urea precipitation method, at 600 oC for sol-gel combustion method. Briefly explain the reason for maintaining two different calcination temperatures for two different methods.
Reply: Thank you so much for your kind comments.
Based on the thermal decomposition behavior of the two precursors (as shown in Fig. 1), the precursor synthesized by the sol-gel method has no obvious weight loss after 600 oC, and the precursor synthesized by the urea precipitation method has no obvious weight loss after 850 oC. Meanwhile, we tend to obtain nanocomposite powder at a lower temperature in order to ensure the nanometric range of grains.
Comment 3: In section 2.4, give the standard of measurements for the characterization of Nanopowders using Vickers hardness and other analytical instruments. Also, introduce the references for the standards in the manuscript.
Reply: Thank you so much for your kind comments.
We have introduced the reference for the standards in the manuscript.
Comment 4: What is the particle size obtained in both synthesis methods before ball milling? Add the XRD and surface morphology results of freshly prepared nanoparticles using two methods after calcination.
Reply: Thank you so much for your kind comments.
We apologized that we did not pay attention to the particle size of the powder before ball milling, which is our negligence. Due to the COVID-19 epidemic in Shenyang, the use of colleges or testing institutions is restricted, so we can't test the particle size of the nanopowders before ball milling. When the epidemic situation improves, we will adopt your opinions to analyze the particle size of the nanopowders before ball milling. Once again, I would like to express our apologies and your help for our paper.
Comment 5: Explain the effect of the ball milling process on the particle's grain refinement using SEM images.
Reply: Thank you so much for your kind comments.
It is a very good question. Indeed, ball milling can break up the agglomeration of nanopowders, thus improving the nanopowder sintering property. The effect of ball milling parameters, such as ball milling time, the diameter of grinding balls, etc on the nanopowder property will be investigated in our following paper.

Round 2
Reviewer 1 Report
Thank you so much for your answers. In general, the questions about clarity in the introduction and experimental section were fulfilled.
Related to results comments and questions in lines 151 to 162 were not successfully answered. The authors argue that the comments will be replied to in the next papers, but I consider that will be resolved in the present paper. Neither Gibbs free energy nor thermal decomposition of precursor analysis was taken into account. Under this situation, I prefer the next paper with all the results and thermodynamical discussion. This similar argument is depicted in question 7.
In this same question (4) Fig 2. There was no answer about SAED images and analysis to validate the XRD result about the crystallinity of nanopowders. If researchers have the foils, they can add the SAED patterns.
Finally, comment 7. I accept your apologies; however, I consider that a rigorous analysis of mechanical properties must be done, even more, when polycrystalline transparent ceramics have great potential in the application of infrared windows because of their excellent optical and mechanical properties.
In conclusion, because of the answers given by the authors, I consider that in the present form the article can not be published.
Author Response
Dear Editor and Reviewer:
Thank you very much for your letter and for the reviewer’s comments concerning our manuscript entitled “Synthesis of MgO coating Gd2O3nanopowders for consolidating Gd2O3-MgO nanocomposite with homogenous phase domain distribution and high mid-infrared transparency” (ID: Coatings-1885649). These comments are all valuable and very helpful for revising and improving our paper. We have studied the comments carefully and have made corrections which we hope meet with approval for the publication of our paper. Revised portion are marked in red in the paper. The main corrections in the paper and the responds to the reviewer’s comments are as following:
Comment 1:Related to results comments and questions in lines 151 to 162 were not successfully answered. The authors argue that the comments will be replied to in the next papers, but I consider that will be resolved in the present paper. Neither Gibbs free energy nor thermal decomposition of precursor analysis was taken into account.
Reply: Thank you so much for your kind comments.
The Gibbs free energy is indeed important for these reactions. However, the intermediate reaction products cannot be accurately determined. For example, in sol-gel process, the second and third steps of thermal decomposition of the precursor may be as follows [1,2]:
M(C6H8O7)x+NO3-→R-COO-M+R’+COx+NOx+O2+H2O (second step)
R-COO-M+R+NO3-→M-R”+CO2+NO2+ O2+H2O (third step)
Where M stands for metal ion, R represents carboxylate organic group,R’denotes residual organic group, R”is organic group in product.
Therefore,the reactions are complex and the exact Gibbs free energy is difficult to accurately calculate.
In our work, the analysis of thermal decomposition process of precursors is mainly for two reasons. One is to determine the minimum calcination temperature of the precursor according to the decomposition temperature the precursor. It is significant to obtain nanopowders at the lowest calcination temperature to realize the nanometric level of nanocomposites. It can be seen from Fig. 1 that the precursors synthesized by the two methods have no significant weight loss after 600 ℃ and 850 ℃, respectively, so the calcination temperature is determined as 600 ℃ and 850 ℃, respectively. Another is that the different thermal decomposition processes of the precursors synthesized by the two methods may result in different agglomeration states of the nanopowders. It can also be seen from the results in Table 1 and TEM images that the agglomeration of the nanopowders prepared by the sol-gel method is more serious, which may be because the second decomposition process is completed instantaneously, and the reaction process is intense, accompanied by a large amount of heat, thus causing the agglomeration of the nanopowders. For urea precipitation, the entire reaction process is gentle without violent decomposition and abrupt weight loss, having no extra heat required for nanopowder agglomeration. Therefore, the nanopowder synthesized by urea precipitation has lower agglomeration state.
- Blank D.H.A., Kruidhof H., Flokstra J., et al. Preparation of YB2Cu3O7-δby citrate synthesis and pyrolysis. J. Phys. D. 1988, 21, 226-227. https://doi.org/10.1088/0022-3727/1/036.
- Shao Z P, Zhou W, Zhu Z H., et al. Advanced synthesis of materials for intermediate-temperature solid oxide fuel cell. Prog. Mater. Sci. 2012, 57, 804-874. https://doi.org/10.1016/j.pmatsci.2011.08.002.
Comment 2:In this same question (4) Fig 2. There was no answer about SAED images and analysis to validate the XRD result about the crystallinity of nanopowders. If researchers have the foils, they can add the SAED patterns.
Reply: Thank you so much for your kind comments.
We have added the SADP patterns in our paper, the results are consistent with the XRD data.
Comment 3: Finally, comment 7. I accept your apologies; however, I consider that a rigorous analysis of mechanical properties must be done, even more, when polycrystalline transparent ceramics have great potential in the application of infrared windows because of their excellent optical and mechanical properties.
Reply: Thank you so much for your kind comments.
Indeed, the loaded nanoindentation curve can well analyze the mechanical behavior of the samples. However, we regret that our equipment cannot measure and supply the load displacement nanoindentation curve. Currently, the hardness testing method we adopted is the same as that in some literatures, as follow:
Wang J.W., Zhang L.C., Chen D.Y., Jordan E.H., Gell M. Y2O3-MgO-ZrO2infrared transparent ceramic nanocomposites. J. Am. Ceram. Soc. 2012, 95, 1033-1037. https://doi.org/10.1111/j.1551-2916.2011.04928.x.
Xu S.Q., Li J., Li C.Y., Pan Y.B., Guo J.K. Infrared-transparent Y2O3-MgO nanocomposites fabricated by the glucose sol-gel combustion and hot-pressing technique. J. Am. Ceram. Soc. 2015, 98, 2796-2802. https://doi.org/10.1111/jace.13681.
Ma H.J., Jung W.K., Yong S.M., Choi D.H., Kim D.K. Microstructural freezing of highly NIR transparent Y2O3-MgO nanocomposite via pressure-assisted two-step sintering. J. Eur. Ceram. Soc. 2019, 39, 4957-4964. https://doi.org/10.1016/j.jeurceramsoc.2019.07.029.

Reviewer 2 Report
This paper can be accepted for publication.
Author Response
Thank you again for your support and recognition of our work.
